# Food Allergy Risk: A Comprehensive Review of Maternal Interventions for Food Allergy Prevention

**DOI:** 10.3390/nu16071087

**Published:** 2024-04-08

**Authors:** Sara Manti, Francesca Galletta, Chiara Lucia Bencivenga, Irene Bettini, Angela Klain, Elisabetta D’Addio, Francesca Mori, Amelia Licari, Michele Miraglia del Giudice, Cristiana Indolfi

**Affiliations:** 1Pediatric Unit, Department of Human Pathology in Adult and Developmental Age ‘Gaetano Barresi’, University of Messina, 98124 Messina, Italy; saramanti@unime.it (S.M.); francygall.92@gmail.com (F.G.); 2Department of Woman, Child and of General and Specialized Surgery, University of Campania ‘Luigi Vanvitelli’, 80138 Naples, Italy; chiara.bencivenga03@gmail.com (C.L.B.); elisabettadaddio3@gmail.com (E.D.); michele.miragliadelgiudice@unicampania.it (M.M.d.G.); cristianaind@hotmail.com (C.I.); 3Pediatric Unit, IRCCS Azienda Ospedaliera-Universitaria di Bologna, 40138 Bologna, Italy; irene.bettini@gmail.com; 4Allergy Unit, Meyer Children’s Hospital, IRCCS, 50139 Florence, Italy; francesca.mori@unifi.it; 5Department of Clinical, Surgical, Diagnostic, and Pediatric Sciences, University of Pavia, 27100 Pavia, Italy; amelia.licari@unipv.it; 6Pediatric Clinic, Fondazione IRCCS Policlinico San Matteo, 27100 Pavia, Italy

**Keywords:** children, evidence, food allergy, prevention, pregnancy

## Abstract

Food allergy represents a global health problem impacting patients’ and caregivers’ quality of life and contributing to increased healthcare costs. Efforts to identify preventive measures starting from pregnancy have recently intensified. This review aims to provide an overview of the role of maternal factors in food allergy prevention. Several studies indicate that avoiding food allergens during pregnancy does not reduce the risk of developing food allergies. International guidelines unanimously discourage avoidance diets due to potential adverse effects on essential nutrient intake and overall health for both women and children. Research on probiotics and prebiotics during pregnancy as preventive measures is promising, though evidence remains limited. Consequently, guidelines lack specific recommendations for their use in preventing food allergies. Similarly, given the absence of conclusive evidence, it is not possible to formulate definitive conclusions on the supplementation of vitamins, omega-3 fatty acids (*n*-3 PUFAs), and other antioxidant substances. A combination of maternal interventions, breastfeeding, and early introduction of foods to infants can reduce the risk of food allergies in the child. Further studies are needed to clarify the interaction between genetics, immunological pathways, and environmental factors

## 1. Introduction

Food allergy (FA) represents a growing global public health problem affecting up to 10% of the world’s population and has increased significantly over the past two decades, especially in children [1,2,3,4,5]. Epidemiological data are predominantly related to the age, diet, and methods of diagnosis [1,6,7,8]. In industrialized countries, children’s most common FA causes are cow’s milk (CM), eggs, peanuts, tree nuts, wheat, soy, fish, and shellfish [9]. The natural history of FAs varies according to the specific allergen. More than 50% of children with CM and egg allergies reach tolerance between 2 and 10 years of life. Only 10–20% of subjects with nut and tree nut allergies achieve spontaneous clinical tolerance [10]. FAs negatively impact the individual’s health and quality of life, resulting in substantial healthcare costs [11,12,13]. As a result, in recent years, there has been a growing focus on preventing food allergies starting from pregnancy. This review aims to provide an overview of the role of maternal factors in FA prevention. A comprehensive search of the published literature was carried out using the PubMed MEDLINE database. Key terms included “food allergy”, “prevention”, and “pregnancy”. Only English-language studies were considered. High-quality studies, such as randomized controlled studies, observational studies, meta-analyses, reviews, and evidence-based guidelines, were included in our analysis.

## 2. Maternal Factors Influencing FAs in Offspring

### 2.1. Genetic Predisposition, Immunomodulation, and Interaction with Environmental Factors

A family history of FAs is a risk factor for FAs [14]. Several genes have been associated with food sensitization and involved in allergen presentation, the T-helper (Th) 2 immune system, and the skin barrier, such as mutations in the gene encoding filaggrin [15,16]. Although genetic factors contribute to the development of FAs, the rapid increase in its prevalence suggests that epigenetic modifications likely play a role in susceptibility to allergies [17]. It has been hypothesized that environmental exposures during the gestational phase in utero induce epigenetic changes that disrupt the immunological state of food tolerance [17,18,19]. For instance, changes in the DNA methylation status of genes for transcription factors, cytokine expression, antibody production, and T lymphocytes have been identified [17,20,21,22,23]. Immunomodulation is also recognized to play a significant role [24]. It is well established that during pregnancy, decidual tissues downregulate maternal Th1 responses. In contrast, the neonatal immune system is characterized by increased production of Th2 rather than Th1 cytokines; subsequently, postnatal environments modulate the responses with an elevation in Th1 activity and a reduction in Th2 activity [24]. However, this pattern does not occur in newborns who develop allergic diseases, where Th2 activity remains more sustained [24]. In this context, several responses are elicited upon exposure to allergens and environmental factors, which influence allergic predisposition differently [24]. Amniotic fluid contains antigens to which the mother has been exposed, and these are ingested by the fetus, representing a potential pathway for Th2 sensitization in the small intestine [24]. Additionally, the amniotic fluid contains immunoglobulin (Ig)E antibodies equivalent to 10% of maternal circulating levels at 16 weeks of gestation [24,25]. Maternal IgE is accountable for a phenomenon known as “antigen focusing”: maternal IgEs show a high affinity for fetal IgE receptors expressed in the intestine, resulting in a higher risk for sensitization even in low antigen concentrations [26]. Various environmental factors have been deemed responsible for inducing epigenetic changes and influencing immunomodulation. Among these, cigarette smoke appears to be correlated with higher levels of Th2 cytokines in children born to smoking mothers compared to those born to non-smoking mothers [27]. Exposure to polycyclic aromatic hydrocarbons is also theorized to impair immune function and contribute to an increased susceptibility of children to allergic diseases [28,29]. In mouse models, particles from traffic smoke were observed to induce hypermethylation of the IFN-γ promoters and hypomethylation of the IL-4 promoter, both significantly associated with changes in IgE levels [30,31]. Notably, higher concentrations of 1-hydroxypyrene in children’s urine were found to elevate the risk of FAs in children [28]. Other environmental factors hypothesized to influence epigenetic structures include xenobiotic chemicals, endocrine disruptors, heavy metals, and radiation [20]. Several studies have demonstrated a correlation between maternal stress during pregnancy and the development of respiratory allergies, atopic dermatitis, and FAs [32]. Elevated prenatal maternal stress was associated with increased immunoglobulin E levels in cord blood [33].

### 2.2. Role of Maternal Diet and Gut Microbiota in Allergy Development

Increasing evidence suggests that nutrition may influence susceptibility to FAs via epigenetic mechanisms triggered by nutritional factors or by modulating the gut microbiota and their functional products [34,35]. Several studies have demonstrated that a varied and balanced diet contributes to developing a healthy immune system in offspring [34,35]. Conversely, a maternal diet high in sweets and trans fats has recently been linked to an elevated risk of FAs in susceptible children [36]. A diet rich in ultra-processed foods is deficient in nutritional value and contains numerous advanced glycation end products (AGEs), which are implicated in FAs [37]. AGEs specifically bind to receptors for advanced glycation end products (RAGEs), which are associated with signaling tissue damage and potentially facilitating abnormal antigen presentation and the development of food allergies. Moreover, AGEs can activate mast cells, triggering the release of proallergic mediators [38]. Furthermore, maternal malnutrition and the resulting deficiency of nutrients such as folic acid, vitamins, and iron can influence the development of atopic diseases in offspring [39]. Specifically, iron deficiency is known to significantly affect immune cells, promoting the survival of Th2 cells, facilitating the class-switching of immunoglobulins towards IgE production, and triggering mast cell degranulation [40]. The mother’s nutritional status is “inherited” by the fetus, meaning that iron deficiency during pregnancy increases the risk of atopic diseases in children, while both allergic children and adults are more prone to suffering from anemia [40]. Conversely, an improvement in iron levels seems to offer protection against the development of allergies [40]. Additionally, recent studies suggest that the microbiome may play a role in the development of FAs [41,42,43,44,45,46]. The immunomodulatory effect of the gut microbiome may depend on its potential to produce short-chain fatty acids (butyrate, acetate, and propionate), which may protect against the development of food allergies by reducing the production of Th2-interleukins and modulating immune response in the intestine and peripheral tissues [47,48]. Since the microbiota are transferred from the mother to the child, maternal dysbiosis can be reflected in the newborn’s dysbiosis, leading to FA development [49,50]. For instance, infants allergic to CM exhibited an elevated presence of *Ruminococcaceae* and *Lachnospiraceae* in their intestinal microbiota compared to healthy 4-month-old controls of the same age, where *Bifidobacteriaceae*, *Enterobacteriaceae*, and *Enterococceae* were predominant [50] More recently, an increased rate of *Bifidobacterium*, *Faecalibacterium prausnitzii*, and *Akkermansia muciniphila* has been linked to the absence of FAs, while a reduction in *Escherichia coli* has been associated with the development of FAs [49,50]. Additionally, several studies supported that mode of delivery—cesarean section—can negatively affect the onset of allergic disease [49,51,52,53]. Infants born by cesarean section develop a different colonization pattern of the intestinal microbiota as they are not exposed to maternal vaginal microbes [51]. Lower levels of Bacteroides, lower diversity within the phylum Bacteroidetes, and a higher level of diversity within the phylum Firmicutes were found in newborns undergoing cesarean section [51].

## 3. Dietary Modification in Preventing Food Allergy

### 3.1. Allergen Avoidance vs. Exposure

Over the past few years, several studies have extensively examined the controversy regarding avoiding and being exposed to food allergens during pregnancy [54]. The effects of consuming food allergens while pregnant or breastfeeding on the development of food allergies in infants are not always well established. The Food Standards Agency found no evidence that avoiding one or more food allergens, with or without other treatments, was linked to a lower risk of FAs in children [54]. Azad et al. showed that a mother’s consumption of peanuts during pregnancy, followed by the introduction of peanuts into the infant’s diet while still breastfed, can prevent peanut sensitization [55]. In another study, out of all the combinations of mother and newborn peanut intake, direct introduction of peanuts during the first year of life, combined with maternal peanut consumption during breastfeeding, was related to the lowest incidence of peanut sensitization [56]. Venter et al. [57] investigated the impact of dietary diversity during pregnancy and lactation on allergic manifestations in offspring. Mothers completed food propensity questionnaires at two time points during pregnancy [57]. The study found no significant associations between maternal diet and infant FAs [57]. On the other hand, Tuokkola et al. [58] demonstrated that a high consumption of dairy milk products during pregnancy can protect babies, particularly those of nonallergic mothers, from developing CM allergy. Coffee, the most popular beverage globally, possesses anti-inflammatory and antioxidant qualities attributed to its bioactive ingredients. In a prospective study, Tanaka et al. [59] firstly established a positive correlation between maternal caffeine intake during pregnancy and the risk of FAs in children, although the underlying mechanisms remain unclear. In recent years, the healthier Mediterranean diet (MD), characterized by an abundance of fruits and vegetables, whole grains, legumes, nuts, olive oil, and fish, has assumed an important role in FA prevention [60]. A recent systematic review revealed a favorable influence of the MD on the emergence of FAs [60]. This correlation is expected due to the well-established health-promoting and anti-inflammatory characteristics of the MD, which is enriched with essential nutrients such as polyphenols, *n*-3 long-chain PUFAs, and other fat-soluble nutrients [60].

#### Guideline Statement

The guidelines for preventing FAs have changed over the last two decades. According to the recent European Academy of Allergy & Clinical Immunology (EAACI) guidelines [61], avoiding potential food allergens during pregnancy may have a minimal or no effect on the development of FAs in early childhood, but the evidence is highly uncertain. Therefore, avoiding diets containing major allergens is not recommended, and eliminating food groups can reduce the intake of vital nutrients and fiber, negatively affecting the health of women and their children [61]. However, this recommendation needs more support as the number of studies is limited, and the certainty of the evidence is restricted [61]. Likewise, the American Academy of Allergy, Asthma, and Immunology (AAAAI), in collaboration with the American College of Allergy, Asthma, and Immunology (ACAAI) and the Canadian Society for Allergy and Clinical Immunology (CSACI), emphasizes that a maternal exclusion diet is not recommended [62]. Similarly, the American Academy of Pediatrics (AAP) does not endorse maternal dietary elimination as a method of allergy prevention [63]. These same recommendations have also been endorsed by the Japanese Pediatric Guidelines for Food Allergy (JPGFA) [64] and the Australian Society of Clinical Immunology and Allergy (ASCIA) [65]. Guidelines/recommendations are shown in Table 1.

### 3.2. Probiotics and Prebiotics

Probiotics are live and active microorganisms that are ingested to maintain or improve the normal microflora in the body [67]. Instead, prebiotics are indigestible food ingredients that provide a source of nutrition for gut microbiota [67]. Some commensal bacteria can ferment prebiotics into short-chain fatty acids (SCFAs), which impact several cellular, molecular, and immunological functions and might help prevent allergy development [67]. Supplementation of prebiotics or probiotics could mitigate the risk of allergies due to the potential effect of probiotics in altering the gut microbial flora and directing the Th2 (atopic) immune response toward Th1 [67]. However, their effectiveness in preventing FAs is still inconsistent [68,69]. Numerous randomized controlled studies have investigated the impact of maternal probiotic supplementation on preventing FAs, but they have yielded mainly undesirable results [70,71]. According to a study by Selle et al., prebiotic treatment in pregnant mice establishes a tolerant environment and microbial imprint in the offspring, thereby mitigating the development of FAs [49].

#### Guideline Statement

There is no specific recommendation either in favor of or against the use of prebiotics, probiotics, or symbiotics, alone or in combination with other approaches, for preventing FAs in pregnant women [61,62]. Existing guidelines acknowledge an impact on eczema prevention but do not endorse using prebiotic or probiotic supplementation to prevent FAs [61,62,64]. Similarly, the World Allergy Organization (WAO) guidelines do not currently offer a specific recommendation regarding prebiotic supplementation during pregnancy or breastfeeding [66]. However, WAO guidelines highlight the importance of providing evidence-based information to clinicians, healthcare professionals, and parents regarding prebiotics to prevent allergies in healthy, full-term infants [66] (Table 1).

### 3.3. Omega-3 Fatty Acids

Omega-3s (*n*-3 PUFAs) are essential fatty acids characterized by the position of the first double bond that, beginning the count from the terminal carbon (ω carbon), occupies the third position, hence the term omega-3 [72]. They mainly include alpha-linolenic acid (ALA), eicosapentaenoic acid (EPA), and docosahexaenoic acid (DHA) [72]. Numerous research has been conducted on the possibility that diets rich in *n*-3 long-chain polyunsaturated fatty acids (LCPUFA) may influence the onset of IgE-mediated diseases [72]. Palmer et al. suggested omega-3 supplementation to pregnant women at high risk of allergies beginning at 21 weeks’ gestation [73,74]. Although the number of children having egg sensitization (defined as a positive skin prick test or specific IgE) at 12 months of age was lower in the group receiving omega-3 supplementation compared to the control group, there was no difference in FA rate between the two groups up to three years of life [73,74]. In their systematic review, Garcia-Larsen et al. reported reduced sensitization to peanuts and eggs in infants whose mothers received omega-3 fatty acids during pregnancy and/or lactation [54]. Vahdaninia et al. [75] reported similar findings in their systematic review and meta-analysis, suggesting that omega-3 LCPUFA supplementation during pregnancy could potentially reduce the incidence of egg and peanut sensitization. However, the available evidence is restricted due to the limited number of studies included in the meta-analyses [75]. Contrarily, a recent systematic review found no significant difference in the incidence of FAs in offspring when mothers were supplemented with omega-3 fatty acids during pregnancy [76]. A study investigated whether combining fish oil and probiotic supplements during pregnancy affects the risk of allergic diseases in children at 24 months [77]. The primary findings indicated that the intervention with fish oil and/or probiotics during pregnancy did not decrease the risk of FAs or atopic eczema [77].

#### Guideline Statement

While incorporating long-chain PUFAs into the diets of expectant and nursing mothers showed a positive impact on the development of FAs and allergy symptoms in offspring, it is not recommended to rely on maternal omega-3 supplementation as a preventive measure for FAs. This caution is due to uncertain evidence [61,62]. Pregnant women are encouraged to follow a healthy Mediterranean diet, which is crucial for the appropriate development of the fetus, and to supplement any deficiencies regardless of FA prevention [60,61] (Table 1).

### 3.4. Vitamin D

Vitamin D is a fat-soluble vitamin and an essential regulator of calcium and phosphorus balance [78]. Approximately 80% of vitamin D (calciferol) is produced in the skin of most animals, including humans, from its precursor, 7-dehydrocholesterol, through exposure to ultraviolet light from sunlight, while only 20% is introduced via the diet. This process yields a naturally occurring form of the vitamin referred to as vitamin D3 [78]. The worldwide prevalence of vitamin D deficiency remains uncertain, but it tends to be higher among individuals residing at high latitudes, particularly during the winter months when daylight hours are limited. Additionally, individuals with darker skin may experience reduced capacity to produce vitamin D, further exacerbating deficiency, particularly when clothing inhibits UV radiation absorption from sunlight [78].

Vitamin D modulates the immune system by reducing the T cells’ release of inflammatory cytokines, which trigger an allergic reaction, and by encouraging the induction of T-regulatory cells, which promote tolerance [79,80]. Limited research has explored the influence of vitamin D during pregnancy and lactation on offspring’s FA development, yielding conflicting results [81,82,83,84,85]. In a study with nearly 100,000 pregnant women, Shimizu et al. reported no definitive link between vitamin D intake during pregnancy and the occurrence of FAs in infants by the age of one year [81]. Likewise, a recent randomized clinical trial has determined that oral vitamin D supplementation during early infancy does not seem to decrease the onset of allergic diseases in early childhood among children deemed “at high risk of allergy” with adequate vitamin D levels at birth [86]. Vitamin D supplementation as a preventive measure for allergic disorders in expectant mothers is not recommended, as an elevated vitamin D level correlated with an increased early risk of developing allergic diseases [82]. A randomized controlled trial indicated that maternal vitamin D supplementation during breastfeeding might elevate the risk of developing FAs up to age 2 years [83]. A 2015 review suggests that low UVB exposure, inferred from latitude and season of birth, may be associated with an increased risk of FAs, though the precise connection between vitamin D and FAs remains unclear [84]. Tuokkola et al. demonstrated a potential correlation between a lower risk of CM allergy in offspring and the mother’s vitamin D intake from foods during pregnancy [85]. Conversely, an elevated risk of CM allergy has been associated using supplement of both vitamin D and folic acid [85]. A systematic review concluded that there is currently insufficient data to support the use of vitamin D supplementation during pregnancy as a preventive measure for the development of FAs [76]. Nevertheless, encouraging results derive from a randomized clinical trial demonstrating that prenatal supplementation with cholecalciferol has a protective effect on the risk of infantile atopic eczema [87].

#### Guideline Statement

According to the current guidelines, the evidence concerning vitamin supplements is of very low certainty due to the wide variety of supplements, doses, timings, target groups, and intervention combinations employed in the studies [61,62]. Consequently, it is not possible to recommend vitamin D supplementation in pregnant women, breastfeeding mothers, or healthy infants for FA prevention [61,62,63,88]. Nevertheless, if a pregnant woman is deficient in vitamin D, supplementation is deemed necessary, regardless of the goal of preventing FAs [61] (Table 1).

### 3.5. Other Factors Implicated in FA Prevention

Several prevalence studies on large cohorts have highlighted a correlation between iron deficiency anemia and the development of atopy [89,90,91,92]. According to Shaheen et al., routine iron supplementation may reduce the risk of asthma in offspring, potentially offering a primary prevention strategy [93]. Additionally, in a double-blind placebo-controlled pilot study, supplementation with a food for special medical purposes, a “lozenge” containing β-lactoglobulin with iron, polyphenols, retinoic acid, and zinc (holoBLG lozenge), was evaluated in allergic women. The study concluded that it effectively increases labile iron levels in immune cells and reduces symptom burden in allergic women, providing evidence that dietary nutritional supplementation of the immune system is a method to combat atopy [94]. Over the years, particular attention has also been given to the role of other antioxidants, such as β-carotene, vitamin C, zinc, copper, and vitamin E [39]. The intake of vitamin C is closely associated with the absorption of iron, as it enhances the uptake of non-heme iron, the form found in plant-based foods and iron supplements [39]. Along with reducing ferric iron, vitamin C helps maintain non-heme iron in its soluble and more absorbable ferrous form. Additionally, vitamin C may regulate iron transport proteins, aiding in iron absorption by intestinal cells. Overall, vitamin C presence in the diet significantly improves non-heme iron absorption, thereby enhancing overall iron levels in the body [39]. A study investigated the possible association between maternal intake during pregnancy of antioxidants (β-carotene, vitamin C, zinc, copper, vitamin E) and the onset of allergies in infants up to one year old [95]. The findings revealed protective associations between antioxidant consumption during pregnancy and allergic outcomes in infants. However, this was observed only for certain antioxidant factors (such as vitamin C and copper) and solely in relation to dietary intake, not dietary supplements themselves [95]. Specifically, increased maternal dietary intake of copper during pregnancy was linked to a decreased risk of wheezing and the development of any early allergic disease in infants with a family history of allergies. Additionally, higher maternal dietary intake of vitamin C during pregnancy seemed to offer protection against wheezing in the first year of life [95]. In a separate study, Gromadzinska et al. [96] explored a potential correlation between vitamin A and E levels during the first trimester, at delivery, and in cord blood and the onset of allergies in newborns up to the age of two. The study differentiated between infants exposed to cigarette smoke and those who were not. By conducting a multivariate analysis, the authors did not find a statistically significant association between vitamins A and E and the risk of developing FAs in children up to 2 years of age [96]. Additionally, exposure to cigarette smoke did not influence the results [96]. Kusmierek et al. [97] conducted a retrospective study on the dietary habits of pregnant mothers whose unborn children were later diagnosed with CM protein allergy. The study revealed that mothers with allergic children had lower intake of vitamin D, vitamin A, LC-PUFA, retinol, riboflavin, and fish; in contrast, mothers of non-allergic children exhibited higher consumption of beta-carotene and folate in their diets [97]. Notably, β-carotene and retinol are both crucial forms of vitamin A, distinguished by their sources, functions, and bioavailability. β-carotene, present in plant-based foods, acts as a precursor to vitamin A and functions as an antioxidant. In contrast, retinol, found in animal products, serves as the active form of vitamin A and plays vital roles in numerous physiological processes [78]. Additionally, Tuokkola et al. [85] explored the connections between maternal consumption of antioxidant vitamins and minerals through diet or supplements during pregnancy and the subsequent development of CM allergy in offspring. The study revealed an increased risk of CM allergy in children associated with the overall intake of beta-carotene by the mother as well as the consumption of beta-carotene and vitamin E from food sources [85]. Lastly, The EAACI systematic review showed inconclusive evidence from copper and vitamin C supplementation during pregnancy to mitigate the risk of FAs in the offspring [76]. Looking at other environmental factors, a study revealed that having a dog at home before and during each trimester of pregnancy reduced the likelihood of FA onset in the offspring up to the first year of life [98,99]. Conversely, possessing a pet other than a dog, such as a cat, hamster, guinea pig, or rabbit, before conception and during each trimester increased the likelihood of developing FAs during the first year of life of the babies [98,99].

#### Guideline Statement

Current guidelines do not recommend the supplementation of additional antioxidants, including vitamins A, C, and E, β-carotene, zinc, retinol, and copper, due to the lack of evidence [61,62]. Any recommendations regarding other environmental preventive factors, such as household pets, can be formulated [61,62] (Table 1).

## 4. Study Limitations

This study has limitations and aspects that should be considered. Foremost is the increased incidence of migratory bias. Patients born and raised in certain environments and later migrating to different countries may develop new allergic reactions during pregnancy due to exposure to different allergens. An example is provided by the review conducted by Berghi et al., which refers to exposure to Solanum Melongema in Asia and subsequent cross-reactivity with other plants in European settings during pregnancy [100]. Another aspect to consider is that there are numerous studies on this topic in animal models, but at present, the translation of these studies to humans is not available.

## 5. Future Research and Conclusions

Several clinical trials regarding the prevention of FAs during pregnancy are currently ongoing. The PrEggNut study investigates whether regular consumption of a high egg- and peanut-containing diet in pregnancy can prevent the development of challenge-proven FAs at 12 months of age (ACTRN12618000937213) [101]. The SYMBA study aims to assess if high-fiber/prebiotic supplements introduced from the second trimester of pregnancy to six months post-partum can reduce the risk of infant allergic disease (ACTRN 12615001075572) [102]. A further study investigates whether the prescription of a personalized diet aimed at enhancing gut colonization of *Prevotella* sp. and butyrate levels in pregnant mothers (NCT04885959) can negatively affect the incidence of allergic diseases in infants [103]. Additionally, The National Institute of Allergy and Infectious Diseases (NIAID) is currently conducting a prospective pilot study to investigate whether early exposure to the mother’s vaginal microbiome in newborns delivered by cesarean section can impact the rate of food allergen sensitization by the end of the first year of life (NCT03567707) [104]. However, due to the lack of evidence, no definitive conclusion can be reached [61,62,63,64]. Moreover, the variability in recommendations may be attributed to differences in study populations, exposure times, doses, outcomes, and the types of interventions employed [95]. It is reasonable to hypothesize that a combination of maternal interventions, breastfeeding, and early introduction of foods in the infant diet could reduce the risk of FAs in the child. Further studies are needed to clarify the interaction between genetics, immunological pathways, and environmental factors.

## Figures and Tables

**Table 1 nutrients-16-01087-t001:** List of the current recommendations for FA prevention.

**Intervention: Allergen elimination diet during pregnancy**		
*Guidelines*	*Recommendation*	*References*
EAACI *	Not recommended	[61]
AAAAI/ACAAI/CSACI *	Not recommended	[62]
AAP *	Not recommended	[63]
JPGFA *	Not recommended	[64]
ASCIA *	Not recommended	[65]
**Intervention: Supplementation of probiotics and prebiotics during pregnancy**		
*Guidelines*	*Recommendation*	*References*
EAACI	No recommendations for or against	[61]
AAAAI/ACAAI/CSACI	Not recommended	[62]
JPGFA	Not recommended	[64]
WAO *	Not recommended	[66]
**Intervention: Supplementation of omega-3 fatty acids during pregnancy**		
*Guidelines*	*Recommendation*	*References*
EAACI	Not recommended	[61]
AAAAI/ACAAI/CSACI	Not recommended	[62]
**Intervention: Supplementation of vitamin D during pregnancy**		
*Guidelines*	*Recommendation*	*References*
EAACI	Not recommended	[61]
AAAAI/ACAAI/CSACI	Not recommended	[62]
**Intervention: vitamins A, C, and E, β-carotene, zinc, retinol, and copper**		
*Guidelines*	*Recommendation*	*References*
EAACI	Not recommended	[61]
AAAAI/ACAAI/CSACI	Not recommended	[62]

* EAACI: European Academy of Allergy & Clinical Immunology; AAAAI: American Academy of Allergy, Asthma, and Immunology; ACAAI: American College of Allergy, Asthma, and Immunology; CSACI: Canadian Society for Allergy and Clinical Immunology; AAP: American Academy of Pediatrics; JPGFA: Japanese Pediatric Guidelines for Food Allergy; ASCIA: Australian Society of Clinical Immunology and Allergy; WAO: World Allergy Organization.

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
