# Peer review of "Food Allergy Risk: A Comprehensive Review of Maternal Interventions for Food Allergy Prevention"

_nutrients, 2024, doi:10.3390/nu16071087_

Round 1

Reviewer 1 Report

Comments and Suggestions for Authors

This is a nice short review, that though could benefit in detailing the mechanism and nutritional aspects in food allergy prevention.

Comments:

Maternal diet has a profound impact on shaping the immune system, in which malnutrition per se (and here particularly iron-deficiency) primes the system towards Th2.( PMID: 25153531, PMID: 35769558)

As such a greater emphasis and mechanistic exlaination should be implemented in section 2.   

3.2. the definition on prebiotics is not entirely correct.

Prebiotics are not nutrients that result from gut microbiota degradation, but they serve as food source for the gut microbiota, ,,,,,,

3.4. It is well established, that Vitamin D is predominantly produced in the skin and in fact only  about 20% is consumed via the diet (PMID: 36698466), but also that calcium is needed to render VitD in its active form. Some explanatory words in regard, why this put some patients groups at greater risk  to develop Vitamin D deficiencies are  welcomed as well as the mentioning of other clinical trials  as PMID: 35763390,.PMID: 32545250 on this subject.

Some studies on Iron-supplementation trials and allergy outcome should be implemented as PMID: 32139461, PMID: 35263681, because particulare here  deficiencies are associated in large trials with the allergic outcome PMID: 26619045, PMID: 32197291, PMID: 31760939, PMID: 36297019.

3.5.

Vitamin C intake is closely linked with iron-bioavailability. It would be nice to interpret in  few words the outcome of the trials.

The authors should detail the differences in ß-carotenes and retinol, particularly in regard to its bioavailiability to better understand the different findings in th Kusmierek -study

Author Response

Dear Reviewer,

Thank you for your review of our manuscript entitled “Food Allergy Risk: A Comprehensive Review on Maternal Interventions for Food Allergy Prevention”. We greatly appreciate the constructive comments. We have read your comments and those of the reviewers and have revised the manuscript. Below, I would like to outline our responses to these comments.

We used point-to-point response to reviewers’ comments. In addition, you will find, coloured in red, suggested modifications in the text.

Thank you again for your interest in our work. We hope that this revision meets with your approval. We await your review of our revised manuscript.

-Maternal diet has a profound impact on shaping the immune system, in which malnutrition per se (and here particularly iron-deficiency) primes the system towards Th2.( PMID: 25153531, PMID: 35769558). As such a greater emphasis and mechanistic exlaination should be implemented in section 2.

Thank you for this insightful suggestion. We have incorporated the requested information into section 2 and cited these two references accordingly (lines 104-113) (ref. 39-40)

-3.2 the definition on prebiotics is not entirely correct. Prebiotics are not nutrients that result from gut microbiota degradation, but they serve as food source for the gut microbiota,

Thank you for this clarification; we amended as suggested (lines 188-189).

3.4. It is well established, that Vitamin D is predominantly produced in the skin and in fact only about 20% is consumed via the diet (PMID: 36698466), but also that calcium is needed to render VitD in its active form. Some explanatory words in regard, why this put some patients groups at greater risk  to develop Vitamin D deficiencies are  welcomed as well as the mentioning of other clinical trials  as PMID: 35763390,.PMID: 32545250 on this subject.

Thank you for this comment. We have incorporated the requested information, referencing the suggested studies (lines 243-251; 258-261;274-276) (ref.81,89,90)

Some studies on Iron-supplementation trials and allergy outcome should be implemented as PMID: 32139461, PMID: 35263681, because particular here deficiencies are associated in large trials with the allergic outcome PMID: 26619045, PMID: 32197291, PMID: 31760939, PMID: 36297019.

Thank you for this valuable suggestion. We have incorporated this topic, adding the required bibliography (lines 289-297) (ref. 92-98)

3.5.Vitamin C intake is closely linked with iron-bioavailability. It would be nice to interpret in few words the outcome of the trials.The authors should detail the differences in ß-carotenes and retinol, particularly in regard to its bioavailiability to better understand the different findings in th Kusmierek -study.

Thank you for this feedback. We have incorporated the requested details to make the presented studies more understandable (lines 289-306; lines 308-316; lines 328-332).

We are grateful to you for the truly helpful comments. These changes will improve the quality of our paper.

Sincerely yours,

Dr. Angela Klain, Department of Woman, Child and General and Specialized Surgery, University of Campania ‘Luigi Vanvitelli’, Naples, Italy.

Reviewer 2 Report

Comments and Suggestions for Authors

Dear Authors,

I read with great interest your article about the possibility of preventing allergies starting from the maternal-fetus interactions.

However, there are some aspects that require your attention.

There are numerous abbreviations in the text and you need to insert a list of abbreviations at the end of the manuscript in order to increase its readability.

You need to include a section of limitations to the present study. One possible aspect that is increasing in incidence is the migration bias. There are patients born and raised in certain environments and afterwards migrate to different countries. There exposure to different allergens can trigger reactions in the proximity of the pregnancy. Such an example could be the exposure to Solanum Melongema in Asia and afterwards cross-reactivity to other plants in European space during pregnancy. Reference this to the article by Berghi ON, Vrinceanu D, Cergan R, Dumitru M, Costache A. Solanum melongena allergy (A comprehensive review). Exp Ther Med. 2021;22(4):1061. doi:10.3892/etm.2021.10495

One other aspect is that there are numerous studies on animal models, but the translation of these studies on humans is not available currently. This should also be mentioned in the limitations section.

Looking forward to receiving the improved version of your manuscript.

Author Response

Thank you for your review of our manuscript entitled “Food Allergy Risk: A Comprehensive Review on Maternal Interventions for Food Allergy Prevention”. We greatly appreciate the constructive comments. We have read your comments and those of the reviewers and have revised the manuscript. Below, I would like to outline our responses to these comments.

We used point-to-point response to reviewers’ comments. In addition, you will find, coloured in red, suggested modifications in the text.

Thank you again for your interest in our work. We hope that this revision meets with your approval. We await your review of our revised manuscript.

-Dear Authors,I read with great interest your article about the possibility of preventing allergies starting from the maternal-fetus interactions. However, there are some aspects that require your attention.There are numerous abbreviations in the text and you need to insert a list of abbreviations at the end of the manuscript in order to increase its readability.

Amended

-You need to include a section of limitations to the present study. One possible aspect that is increasing in incidence is the migration bias. There are patients born and raised in certain environments and afterwards migrate to different countries. There exposure to different allergens can trigger reactions in the proximity of the pregnancy. Such an example could be the exposure to Solanum Melongema in Asia and afterwards cross-reactivity to other plants in European space during pregnancy. Reference this to the article by Berghi ON, Vrinceanu D, Cergan R, Dumitru M, Costache A. Solanum melongena allergy (A comprehensive review). Exp Ther Med. 2021;22(4):1061. doi:10.3892/etm.2021.10495. One other aspect is that there are numerous studies on animal models, but the translation of these studies on humans is not available currently. This should also be mentioned in the limitations section. Looking forward to receiving the improved version of your manuscript.

Thank you for this valuable and appreciated suggestion. We have added a "Study Limitations" section where we included your suggestion, citing the mentioned study (lines 353-360, ref.103)

We are grateful to you for the truly helpful comments. These changes will improve the quality of our paper.

Sincerely yours,

Dr. Angela Klain, Department of Woman, Child and General and Specialized Surgery, University of Campania ‘Luigi Vanvitelli’, Naples, Italy.

Round 2

Reviewer 1 Report

Comments and Suggestions for Authors

I thank the authors to address all my queries.